# Long-Term Residential Environment Exposure and Subjective Wellbeing in Later Life in Guangzhou, China: Moderated by Residential Mobility History

**DOI:** 10.3390/ijerph192013081

**Published:** 2022-10-12

**Authors:** Lingling Su, Suhong Zhou

**Affiliations:** 1Key Research Institute of Yellow River Civilization and Sustainable Development & Collaborative Innovation, Center on Yellow River Civilization, Henan University, Kaifeng 475001, China; 2School of Geography and Planning, Sun Yat-sen University, Guangzhou 510275, China; 3Guangdong Provincial Engineering Research Center for Public Security and Disaster, Guangzhou 510275, China

**Keywords:** residential environment, subjective wellbeing, residential mobility, China, Guangzhou

## Abstract

With rapid global urbanization, the importance of understanding relationships between the changing environment and wellbeing is being increasingly recognized. However, there is still a lack of understanding of how long-term residential environment exposure affects subjective wellbeing under the dual changes of geographical environment and residential location. Based on a survey of the elderly (people over 60 years old) in Guangzhou, China, this study analyzes the effect of long-term residential environment exposure over 25 years on subjective wellbeing in later life in the context of residential mobility. The study found that subjective wellbeing in later life is not only related to the current residential environment but also the cumulative exposure to the long-term residential environment. The relationship between long-term residential environment exposure and subjective wellbeing in later life tends to be stable with the increase of cumulative time, especially the cumulative years over 15 years. Considering the importance of residential mobility history, the study further analyzes the moderating effects of relocation frequency and residential location. Relocation frequency can strengthen the positive effect of residential environment on subjective wellbeing and weaken the negative effect of residential environment on subjective wellbeing, which confirms the existence of residential self-selection. In addition, the direction of effect of residential environment on residents who move between living in the urban center and the periphery is consistent with that of residents who have always lived in the urban center, while the effects of the residential environment on residents who have always lived in the urban center and those who have always lived in the urban periphery are related in different directions. The conclusion of this study can provide guidance for individuals’ residential choice and governance of the urban environment to improve wellbeing.

## 1. Introduction

With the emergence of social models of health, the attention of health geographers has shifted from medically centered concepts of the body and disease to broader concepts of health experienced by individuals [1]. Recently, this attention has greatly promoted the study of subjective wellbeing, partly due to the long-standing interest of health geography in quality of life [2]. Considerable work has been undertaken by health geographers, for whom subjective wellbeing provides a useful concept to move beyond biomedical understandings of health [2,3]. Some scholars believe that subjective wellbeing can enable people to have a more comprehensive and in-depth understanding of health and obtain a broader social experience [4]. In contrast to the traditional view of health as the absence of disease, subjective wellbeing emphasizes the socio-spatial relationship and environmental context of life experiences.

The value of linking the urban environment and wellbeing outcomes is being increasingly recognized; however, the myriad relationships are far from being understood scientifically [5,6]. The effects of residential environment exposure on subjective wellbeing in multiple temporal and spatial dimensions are conditioned by complicated interactions [7]. Both the duration and spatial location of environmental exposure may lead to different subjective wellbeing outcomes [8,9]. Furthermore, residents’ living conditions and environment have persistent differences, which has severe effects on individuals’ subjective wellbeing. To a certain extent, residential mobility is the major cause of continuous population differentiation in environmental health and wellbeing [10]. Existing studies tend to focus on the impact of the current normalized residential neighborhood environment on subjective wellbeing, but for most residents, the residential environment is not static, and especially in recent years, residential mobility has become more and more common [11]. Subjective wellbeing as an overall assessment of a person’s long-term quality of life. It may be important to analyze the relationship between long-term residential environment exposure and subjective wellbeing in the context of residential mobility.

Health geography research based on individual spatiotemporal behavior becomes the key to understanding the urban environment and individual quality of life [12]. Spatiotemporal behavior research focuses on the interaction between humans and spatial environments, analyzes the space behavior of individuals in different spatiotemporal conditions, strengthens the cognition of the correlation between “individuals, time and space”, and provides a new perspective for the complex relationship between environment and subjective wellbeing [13]. In addition, spatiotemporal behavior research focuses on the hierarchical decision-making and selection process from the social group to the family to the individual for both long-term decision making and short-term arrangements, which can interpret the interaction between urban geographic environment and residents’ subjective wellbeing from different temporal and spatial dimensions [14]. The geographical study of subjective wellbeing should not only pay attention to the dynamic changes of geographical environment but also pay attention to the temporal and spatial characteristics of individuals experiencing these environments and thus guide urban governance to improve subjective wellbeing.

In the process of rapid urbanization in China, the urban environment has undergone tremendous changes, and the spatial differences have continued to increase. At the same time, since the housing market reform, residential mobility has become more and more common [15]. Individuals’ subjective wellbeing is not only affected by their current residential environment but also by their long-term history of residential mobility [16]. However, there is still a lack of research on the impact of long-term residential environment exposure on subjective wellbeing. From the perspective of residential mobility, this study focuses on the impact of long-term residential environment exposure on subjective wellbeing in later life and the possible moderating effect of residential mobility history.

## 2. Theoretical Framework

### 2.1. Institutional Reform and Residential Mobility

Residential mobility has become more and more common, especially intraurban residential mobility [17,18]. Unlike long-distance residential mobility for higher-value goals (such as education and employment), intraurban residential mobility is seen as driven by household transitions or dwelling and neighborhood preferences [19,20]. The life-course perspective emphasizes that individual lives are embedded within webs that stretch across space and time [21]. Individual experiences are influenced by the macro-contexts. The collective experiences created by these structural forces are known as period and cohort effects [21,22]. Period effects are felt by anyone living in a particular time and place. They live in the same social environment, and the decision-making of residential mobility is affected by the macro-economic environment (such as housing policy, urban renewal or expansion, etc.). Cohort effects refer to the commonalities of experience shared by individuals who are born at the same time and live out their lives under similar structural conditions.

Residential mobility is affected by the broader social and economic environment, such as housing policy and the real estate market [23]. During the period of the planned economy, China’s urban housing was mainly provided by state-owned enterprises and units. As a type of welfare, housing was provided to employees in the form of unit housing, forming a “unit system” housing pattern, and people had fewer opportunities for residential mobility [24,25]. In 1992, when China established a market economy, urban housing reform was launched nationwide, gradually forming a dual-track housing supply system with both public and commercial housing, allowing people to buy or rent housing from the market [26]. In 1998, the Chinese government officially announced the end of the welfare housing allocation system and the beginning of the marketisation of housing, which greatly improved the residential mobility rate [27,28].

### 2.2. Residential Environment and Subjective Wellbeing 

As a material entity, residential space supports people’s daily life, and the residential environment is the foundation of subjective wellbeing. An individual’s subjective wellbeing may be the product of decades of environmental exposure. Previous studies on residential environment rarely considered the variability and long-term nature of the residential environment in the life course and focused on the relationship between residential environment and subjective wellbeing in the same time period without considering that living environment in the early stage of the life course will affect subjective wellbeing in later life. Studies have pointed out that long-term environmental exposure is more important than current exposure because the residential environment may have a continuous impact over time [10,29,30]. Residents living in disadvantaged neighborhoods for a long time may face greater stress and anxiety, which can reduce the quality of life over time [31]. Long-term exposure to the natural environment can promote the establishment of social ties, encourage physical activity, and improve health and wellbeing [32,33].

There is a growing recognition that the residential environment is not static but changes dynamically over time [34]. Over the life course, people experience a variety of situations as they move in and out of different residential environments [35,36]. Even in a single place of residence, the physical and social environment may change substantially over time with spatial facilities and social development [34]. Therefore, limiting the impact of residential environment to a single point in time and ignoring the accumulated experiences of people in different residential settings may lead to a false assessment of the influence of geographical environment on subjective wellbeing.

Most of the existing studies have examined the short-term effects of residential environment but ignored the complexity of long-term residential environment on subjective wellbeing in later life. The accumulation of early life experiences may have an impact on later life, especially in old age. Population aging is a global social phenomenon. The elderly in China, defined as over 60 years old, have experienced significant social transformation in their lifetime, and their residential environments and choices are affected by their specific social backgrounds.

### 2.3. Residential Mobility History

Although residential context has profound effects on individuals’ subjective wellbeing, people can change their subjective wellbeing by moving to a new location. Research on the effects of long-term residential environment exposure on subjective wellbeing cannot ignore people’s life experience of moving between different residential environments. Traditional cross-sectional research does not distinguish between the short-term impact of the residential environment (for movers) and the long-term impact (for stayers), which may lead to biased results. Therefore, the relationship between long-term residential environment exposure and subjective wellbeing needs to consider the impact of residential mobility history.

Residential mobility has generally been viewed as a negative and stressful event, which is substantiated by numerous studies [36]. People who move frequently are more likely to lose strong social ties and networks, and it is associated with various health risks [37,38]. Frequent mobility may also lead to unobserved influences, such as family instability, environmental sensitivity, etc., which may be important potential factors affecting subjective wellbeing. Some scholars, from the perspective of people moving to opportunity, believe that residential mobility may have positive effects because residential mobility allows people to leave unfavorable living conditions and move to better environments [39]. Seeking better opportunities has promoted the residential mobility of people from the suburbs to the inner city.

In China’s big cities, there are significant spatial differences in the urban environment. The central area of a city is a high-density area of population activity, while the peripheral area is in a relatively closed state with less social differentiation [40]. There are great differences between the central and peripheral areas of a city in terms of population socioeconomic attributes, material facilities, community management services, neighborhood network, housing prices, and spatial layout. People living in a specific regional environment for a long time may develop specific behavioral patterns and environmental coping habits. If a resident lives in a very good environment for a long time, that resident may feel uncomfortable once they move to a place where the environment is not so good, while a resident who lives in a normal environment for a long time may have a higher tolerance for that residential environment [41]. Therefore, different residential mobility history (relocation frequency and location) may affect the relationship between residential environment and subjective wellbeing.

### 2.4. Conceptual Framework and Research Hypothesis

Based on the analysis and review above, this paper proposes a conceptual framework to examine and compare the relationship of long-term residential environment exposure and subjective wellbeing under residential mobility (Figure 1). There are two hypotheses. 

**Hypothesis** **1.**
*Subjective wellbeing in later life is related to the long-term residential environment exposure.*


**Hypothesis** **2.**
*This relationship may be moderated by residential mobility history such as relocation frequency and residential location.*


## 3. Data and Methods

### 3.1. Survey Participants

Guangzhou, located in the south of China, is a mega city. The socialist market economic system implemented in 1992 further accelerated the city’s urban expansion and urbanization movement. The built-up area of Guangzhou increased six-fold in 25 years from 206 km^2^ in 1992 to 1249 km^2^ in 2016, and the urbanization rate increased from 69.40% in 1992 to 86.06% in 2016. The urban space has undergone large-scale expansion and reconstruction in a short period of time, resulting in dramatic changes in the characteristics of the living environment and continuous reorganization of the social structure, which has a profound impact on residents’ lives.

The data of this study are based on a questionnaire survey of the elderly conducted in Guangzhou from October to December 2016. We adopted stratified random sampling survey method. Firstly, based on the data of the sixth Census of Guangzhou, the neighborhoods were clustered, and then, a total of 46 neighborhoods were selected from various types of neighborhoods as sample neighborhoods to conduct the random sampling survey of the elderly. More details of the survey are in Su et al. (2021) [42]. A total of 1012 valid questionnaires were collected, and 782 elderly people who have lived in Guangzhou for more than 25 years (since at least 1992) were selected as the sample residents of this study (Table 1). The questionnaire gathered individual attributes, characteristics, and subjective wellbeing information. The questionnaire also recorded the detailed residential history of each person since birth, including relocation time and address.

### 3.2. Subjective Wellbeing

Subjective wellbeing is an individual’s self-evaluation of their own life quality. In colloquial terms, it can simply be considered self-appraisal of whether one is happy or not [43]. At present, the measure of subjective wellbeing mostly adopts self-report; that is, individuals score their subjective wellbeing through questionnaires. In this sense, it takes into account the subjective aspects of preference satisfaction, as it allows people to perceive the quality of their own lives without requiring others to assess their subjective wellbeing [44]. In this study, subjective wellbeing was captured by the survey question “Taken all together, how happy do you feel about your current life?” and measured on a 5-point scale, with 1 representing “very unhappy” and 5 representing “very happy”. The values from 1 to 5 represent increasing levels of subjective wellbeing. Existing studies have shown single items to measure subjective wellbeing are reliable, effective, and comparable [45,46].

### 3.3. Exposure Assessment of Residential Environment 

Residential environment, in this study, is the social and physical environment of the residential neighborhood. Based on literature review and data availability, the population density and the proportion of migrants and a highly educated population (undergraduate and above) in the neighborhood are used as proxies for residential social environment. These variables of residential social environment are derived from data of the Fourth Census (1990), the Fifth Census (2000), and the Sixth Census (2010) of Guangzhou. The neighborhood boundaries in this study are defined based on administrative units created by each census. For physical environment variables, satellite images of Guangzhou were obtained through the Geospatial Data Cloud Platform of the Computer Network Information Center of the Chinese Academy of Sciences, with a spatial resolution of 30 m. Image data with less cloud cover were selected for analysis, and images with higher cloud cover were selected from adjacent years for replacement. To represent the residential physical environment, the normalized difference vegetation index (NDVI), the normalized difference water index (NDWI), and the normalized difference built-up index (NDBI) were obtained through band calculation. NDVI is a numerical indicator that uses the visible and near-infrared bands of the electromagnetic spectrum, which can effectively capture neighborhood green space [47]. NDWI was developed to extract open-water features or blue space and enhance their presence in remote sensing images based on reflected near-infrared radiation and visible green light [48]. NDBI uses the difference in reflection characteristics of buildings in mid-infrared and near-infrared to discriminate impervious surfaces and then extract urban building density [49]. The physical environment variables in this study were obtained based on the 1 km buffer distance surrounding residential location, as this distance has been found to be efficacious in health and wellbeing studies employing multiple buffer distances [50,51].

Assessment of long-term residential environment exposure was based on residential location over the past 25 years. The questionnaire recorded each participant’s residential address and residence time (start/end), which allowed us to accumulate the residential environment variables year by year so that the dual changes of residence and environment could be taken into account at the same time.

### 3.4. Residential Mobility History

The survey obtained the residential mobility trajectory of the elderly in the sample from 1992 to 2016, including the time and detailed address of each relocation. The number of times of residential mobility and residential location during this period were used to analyze the impact of residential mobility history on the relationship between long-term residential environment exposure and elderly subjective wellbeing. Taking the outer ring road of Guangzhou as the boundary, within the ring road is regarded as the central area, and outside the ring road is regarded as the peripheral area. According to the residential location, the sample of elderly people was divided into three types: those who have always lived in the urban center, those who have always lived in the urban periphery, and those who have moved between living in the center and the periphery. 

In addition, gender, age, education, marital status, and self-rated health have been widely discussed in the study of subjective wellbeing [52,53]. Therefore, these variables are included in the model as control variables in this study.

### 3.5. Statistical Analysis

The dependent variable is the current subjective wellbeing, which is a five-category ordinal variable, so the ordered logistic regression model (Ologit) was used to study the association between long-term residential environment exposure and subjective wellbeing at the individual level. The Ologit models are specified as follows:(1)$$P_{j}=P\left(y\leq j|X\right)=\frac{exp\left(a_{j}+\sum_{i=1}^{n}\beta_{i}x_{i}\right)}{1+exp\left(a_{j}+\sum_{i=1}^{n}\beta_{i}x_{i}\right)}$$
where y represents the value of subjective wellbeing; $a_{j}$ represents the regression coefficient of constant term; $\beta_{i}$ represents the coefficient; $x_{i}$ represents independent variables and control variables in the model.
(2)$$\log\left(\frac{P_{j}}{1-P_{j}}\right)=a_{j}+\sum_{i=1}^{n}\beta_{i}x_{i}$$

This equation is the occurrence probability of events in the Ologit model, and *P*/1 − *P* is the occurrence ratio, that is, the ratio of the occurrence probability of the event (*y* ≤ *j*) to the non-occurrence probability of the event (*y* > *j*). The odds ratio (OR) indicates the influence of independent variables on the change in the probability of occurrence of dependent variables.

## 4. Results

### 4.1. Long-Term Residential Environment Exposure and Subjective Wellbeing

The study first constructed the Ologit models of cumulative exposure to the residential environment and elderly subjective wellbeing over the past 25 years (1992 to 2016). Results are presented in Table 2, in which model 1 is the regression result of individual attributes, and model 2 and model 3 add residential social environment variables and residential physical environment variables, respectively.

In the social environment of residential neighborhood, long-term exposure to high population density is significantly related to lower subjective wellbeing in later life (β = −0.453, OR = 0.636, *p* < 0.001). Unlike cities in western countries, many cities in China are characterized by high population density. Large concentrations of population may lead to social disorder and exert pressure on public facilities and housing, which are not conducive to subjective wellbeing [54]. The proportion of highly educated population is significantly positively correlated with subjective wellbeing (β = 0.184, OR = 1.202, *p* < 0.01). This may be because living long term among people with a higher education level is more conducive to owning and accumulating social capital [55]. The proportion of migrants is positively correlated with subjective wellbeing (β = 0.482, OR = 1.620, *p* < 0.001). Since China’s transition to a market economy and the reform of the housing system, residential mobility has become more and more common. In areas with a high proportion of migrants, people obtain social support and job opportunities through friends, acquaintances, and fellow villagers, forming a community with supportive characteristics [56]. An area that can attract migrants for a long time is usually characterized by diversified employment opportunities, excellent public services, and strong cultural inclusiveness [57].

In terms of the physical environment of residential neighborhoods, the elderly with long-term exposure to green space have higher subjective wellbeing (β = 0.352, OR = 1.421, *p* < 0.001). Green space can promote daily physical activities, increase social interaction opportunities, and relieve negative emotions caused by life pressure and thus increase residents’ subjective wellbeing [58,59]. Blue space is significantly negatively correlated with subjective wellbeing in later life (β = −0.186, OR = 0.830, *p* <0.01). This may be because, on the one hand, living for a long time in the humid environment of Guangzhou, where the summer is hot and rainy, and the air is humid, is likely to bring certain health risks, such as asthma, rheumatism, and arthritis [60]. On the other hand, rapid urbanization has brought urban river pollution problems, affecting the quality of life of residents [61]. Building density is significantly negatively correlated with subjective wellbeing (β = −0.215, OR = 0.807, *p* < 0.001). Rapid urbanization and urban sprawl in China have brought high building densities in cities, resulting in smaller and more crowded urban outdoor spaces. At the same time, high building density leads to a series of urban problems such as space congestion, tension in the use of public facilities, and environmental deterioration, reducing subjective wellbeing [62]. Hypothesis 1 is confirmed.

In terms of individual socioeconomic attribute variables, women, married people, and residents with high self-rated health have higher subjective wellbeing, which is consistent with existing research. However, education status is significantly negatively correlated with subjective wellbeing. This may be because the sample of this study is people over 60 years old, and higher educational attainment does not continue to bring higher competitiveness in the labor market. Highly educated people also tend to have higher expectations for social development and their personal life. All these factors may lead to the decline of their subjective wellbeing.

### 4.2. Moderating Effect of Residential Mobility History

China’s rapid urbanization process and housing system reform have brought the differentiation of urban living environments and an increase of residential mobility. The experience of residential mobility in the life course will inevitably have an important impact on the individual’s ability to perceive and adapt to the surrounding environment and living conditions. Therefore, this study analyzes the group heterogeneity of the relationship between the cumulative exposure to long-term residential environment and subjective wellbeing in later life through the frequency and location of residential mobility.

(1) Moderating effect of relocation frequency

Table 3 shows the model results of the moderating effect of relocation frequency on the relationship between long-term residential environment exposure and subjective wellbeing in later life. The interaction coefficient between population density and relocation frequency is significantly positive, indicating that residential mobility weakens the negative impact of long-term exposure to high population density on subjective wellbeing in later life. The interaction coefficient between the proportion of highly educated population and relocation frequency is significantly positive, indicating that residential mobility strengthens the positive impact on subjective wellbeing of living long term in an environment of highly educated people. The interaction coefficient between green space and relocation frequency is significantly positive, indicating that residential mobility strengthens the positive impact on subjective wellbeing of living long term in a green space environment. The interaction coefficient between building density and relocation frequency is significantly positive, indicating that residential mobility weakens the negative impact on subjective wellbeing of long-term exposure to high building density.

In order to more clearly and intuitively understand the moderating effect of relocation frequency on the relationship between long-term residential environment exposure and subjective wellbeing in later life, the study divides relocation frequency into high relocation frequency (2 and above) and low relocation frequency (0 and 1) based on the median and draws a schematic diagram of the moderating effect (see Figure 2). From Figure 2a,d, it can be seen that population density and building density are negatively correlated with subjective wellbeing, but with the increase of population density and building density, subjective wellbeing of residents with low relocation frequency decreases faster. From Figure 2b,c, it can be seen that the proportion of highly educated population and green space are positively correlated with subjective wellbeing, but with the increase of highly educated population and green space, the subjective wellbeing of residents with high relocation frequency can be more effectively improved. This may be because residents with low relocation frequency cannot flexibly adjust their residence and living environment due to economic or family constraints, while residents with high relocation frequency can more flexibly adjust their residence and avoid disadvantages in the choice of residence. Residential mobility can meet the new residential needs and preferences. Especially in recent years, people move so their children can receive a better education and enjoy a better living environment.

(2) Moderating effect of residential location

Residential location is a categorical variable, so the study adopts grouping regression, which overcomes the possible result bias caused by simple regression without distinguishing the types of moderating variables and improves the accuracy of the test.

Table 4 reports the grouping regression results of the moderating effect of residential location since 1992. For the residents who have always lived in the center of Guangzhou, population density, blue space, and building density are significantly negatively correlated with subjective wellbeing in later life, while migrants and green space are significantly positively correlated with subjective wellbeing. For the residents who have moved between living in the center and periphery, population density and blue space are significantly negatively correlated with subjective wellbeing in later life, while a highly educated population, migrants, and green space are significantly positively correlated with subjective wellbeing. For the residents who have always lived in the periphery of Guangzhou, population density, a highly educated population, blue space, and building density are significantly positively correlated with subjective wellbeing in later life, while migrants and green space are significantly negatively correlated with subjective wellbeing. The results show that the relationship between long-term residential environment exposure and subjective wellbeing in later life is affected by the history of residential location.

Figure 3 is a schematic diagram of the moderating effect of residential location. It can be seen more intuitively that there is spatial heterogeneity in the relationship between long-term residential environment exposure and subjective wellbeing in later life. Although there are differences in the intensity of the relationship between residential environment exposure and subjective wellbeing of elderly residents who have always lived in the urban center and those who have moved between the center and periphery, the direction of the effect is relatively consistent. However, the relationship between residential environment exposure and subjective wellbeing is opposite in the residents who have always lived in the urban center and the residents who have always lived in the urban periphery. Hypothesis 2 is confirmed.

### 4.3. Robustness Check

In China’s market economy transformation and housing reform, institutional and market factors play an important role in housing choice and quality of life [63]. Based on this, the study adds relevant institutional factors and market factors as control variables to test the robustness of the conclusions. We selected the type of hukou and the nature of work unit before retirement as institutional factors because they are the product of China’s institutional system. Whether residents have local hukou and the associated entitlements and whether they work in a state-owned enterprise or public institution directly affect the quality of residents’ access to public services such as medical care and education. In addition, personal monthly income and full-time or part-time employment before retirement are selected as market factors because they are closely related to the market economy.

The results show there is no evidence that hukou and work unit are related to the subjective wellbeing of the elderly, while the market factors reached statistical significance (see Table A1 in the Appendix A). The higher the income, the higher the subjective wellbeing, and the subjective wellbeing of the elderly who had full-time jobs before retirement is higher than that of the elderly who had part-time jobs. After adding institutional factors and market factors, the relationship between long-term residential environment exposure and subjective wellbeing in later life did not change significantly, indicating the robustness of the research conclusions.

## 5. Discussion

### 5.1. Accumulated Exposure to Residential Environment

Subjective wellbeing in later life is not only related to current residential environment exposure but also to long-term residential environment cumulative exposure. Cumulative residential environment exposure in the last 24 years, 23 years, 22 years, etc., to the latest year was calculated according to the same method. The odds ratios of residential environment and subjective wellbeing were also computed to better understand the relationship between environment exposure and subjective wellbeing from different cumulative exposure times (Figure 4). The results show that residential environment exposure has continuous positive or negative impacts on subjective wellbeing in later life with cumulative time. In addition, the relationship between cumulative exposure to residential environment and subjective wellbeing in later life fluctuated greatly in the most recent 15 years, especially in exposure to blue space, and tended to be stable with increase of exposure time. Only focusing on the current residential neighborhood environment may overestimate or underestimate the individual effects of geographical environment, and this finding is consistent with previous studies [33].

### 5.2. The Role of Relocation Frequency

The study found that relocation frequency strengthens the positive relationship between long-term residential environment exposure and subjective wellbeing in later life but weakens the negative relationship between them. This may be due to the existence of residential self-selection. According to the theory of residential self-selection, residents choose their residence according to their socioeconomic attributes and attitude preferences [64,65]. Since the housing market reform in 1992, residents have more and more autonomy in their housing choices and can freely buy or rent housing from the market according to their living preferences. People with higher socioeconomic status may have more resources and enjoy better residential options [66]. Through residential mobility, people can avoid the impact of an unfavorable environment and tend to live in a better environment. Appropriate residential mobility can improve residents’ subjective wellbeing. Policy makers and urban managers need to improve and standardize the housing supply system to meet the housing needs of different socioeconomic groups and promote reasonable and healthy residential mobility.

### 5.3. The Role of Residential Location

The relationship between the cumulative exposure of residential environment and subjective wellbeing in later life is affected by the long-term residential location. Residents who live in the urban center and those who have always lived in the urban periphery are affected differently by the geographical environment, which may be due to the differences in environmental habits and perceptions formed over a long period of time. For the residents who have moved between the center and periphery, the relationship between their subjective wellbeing and residential environment is more consistent with the residents who have always lived in the urban center. This may be due to the urbanization movement in recent decades, which has caused residents in the urban peripheral area to move to the central area and move closer to the lifestyle of urban center residents [15]. The geographical environment has spatial heterogeneity, and there are different symbolic dimensions in different locations. For example, the green space in the urban center is mostly cultivated and maintained green space for parks, which has the function of beautifying the environment and providing rest places, while the green space in the urban periphery is mostly farmland and more natural landscapes. Due to the different practices and meanings related to green space, the impact of green space on residents’ subjective wellbeing is also different in the urban center and periphery, which is consistent with the findings in other studies [67,68]. In future urban planning and management, it is necessary to plan according to the functions of different spaces and the needs of the space users to strengthen the livability and sustainability of cities.

## 6. Conclusions

Taking Guangzhou, China, as a case city, this study explores the relationship between long-term residential environment exposure over 25 years and subjective wellbeing in later life and the possible moderating effect of residential mobility history. There are three main conclusions. First, subjective wellbeing in later life is not only related to the current residential environment but also the cumulative exposure to the long-term residential environment. The directions of the relationship between residential environment variables and subjective wellbeing are not affected by cumulative time in the direction (positive or negative) but differ in the intensity. The relationship between long-term residential environment exposure for 15 years or more and subjective wellbeing is relatively stable. Second, the relationship between residential environment and subjective wellbeing is moderated by relocation frequency. Relocation frequency can strengthen the positive relationship between a highly educated population, green space, and subjective wellbeing and weaken the negative relationship between population density, building density, and subjective wellbeing. This is consistent with the theory of residential self-selection. Last, the relationship between long-term residential environment exposure and subjective wellbeing in later life is heterogeneous due to the impact of residential location. Although there are differences in the intensity of effects between residential environment and subjective wellbeing of the elderly between the residents who have always lived in the urban center and those who have moved between the center and periphery, the direction of effects (positive effect or negative effect) is relatively consistent. The subjective wellbeing of residents who have always lived in the urban center and those who have always lived in the urban periphery is related to the residential environment in different directions.

However, the study has some limitations. First, this is a cross-sectional study. Although we consider long-term residential mobility trajectory over 25 years and environmental changes, the causal inference between environment and subjective wellbeing is limited because the individual attributes and subjective wellbeing in this study are cross-sectional data. Thus, future research is needed to examine an explicit causal relationship between geographical environment and subjective wellbeing using a longitudinal design. Second, the study is carried out in the context of China. The elderly in the study have experienced the institutional transformation of the country, which has a specific historical background and unique characteristics. Whether the conclusions of this study are applicable to other countries and regions needs further verification.

## Figures and Tables

**Figure 1 ijerph-19-13081-f001:**
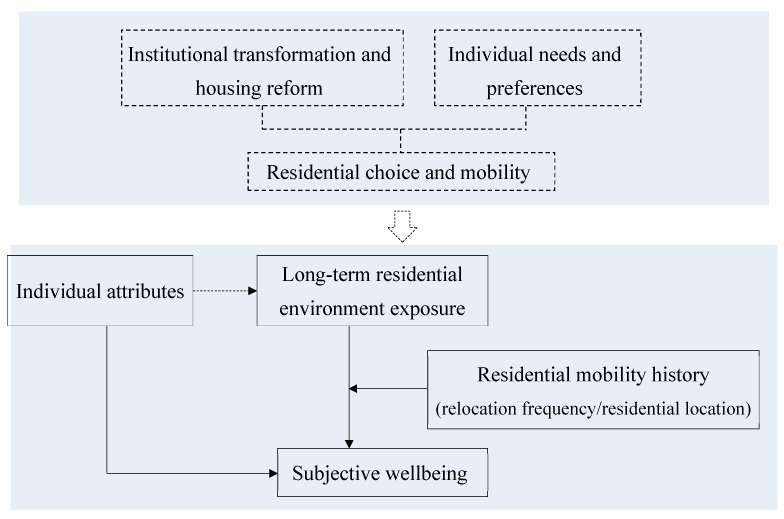
Conceptual framework.

**Figure 2 ijerph-19-13081-f002:**
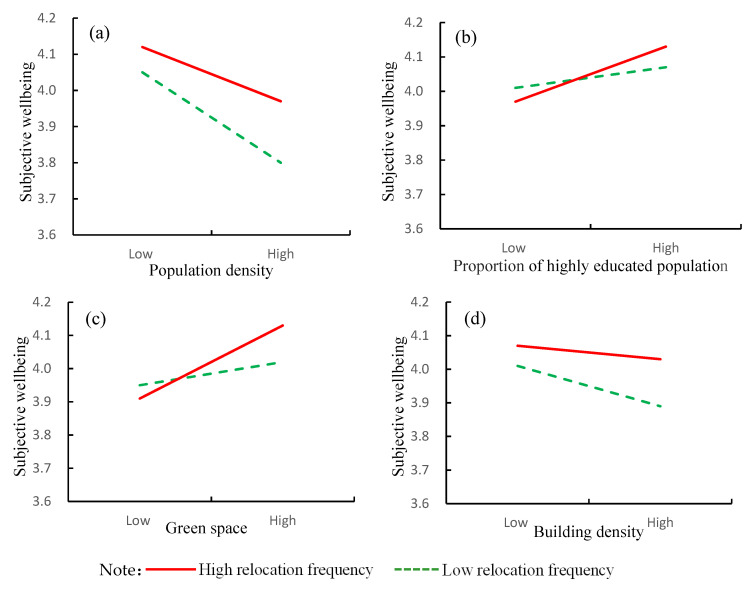
Moderating effect of relocation frequency.

**Figure 3 ijerph-19-13081-f003:**
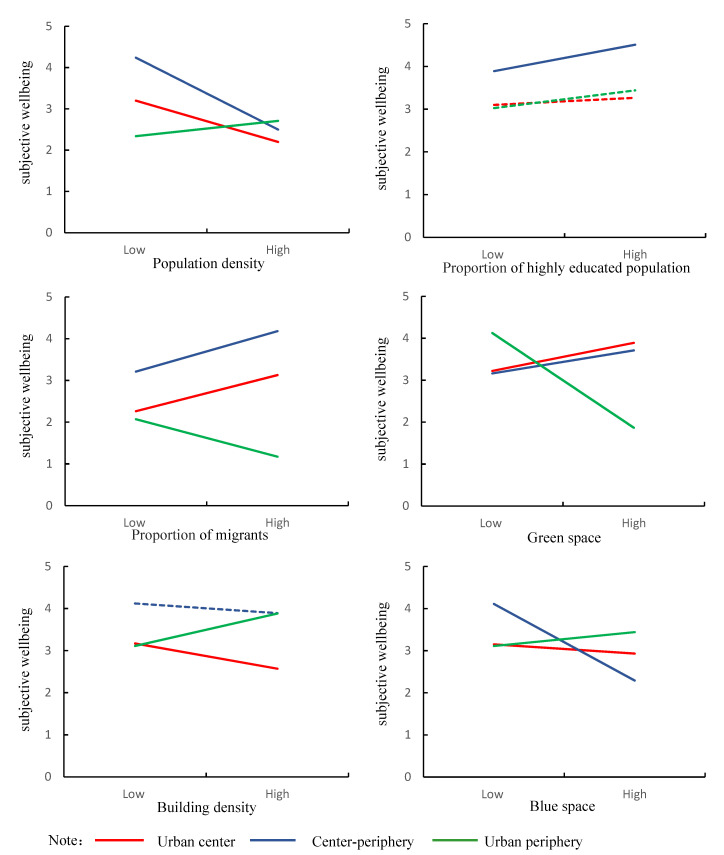
Moderating effect of residential location. Note: Solid lines represent statistical significance at the 90% confidence interval, while dashed lines do not.

**Figure 4 ijerph-19-13081-f004:**
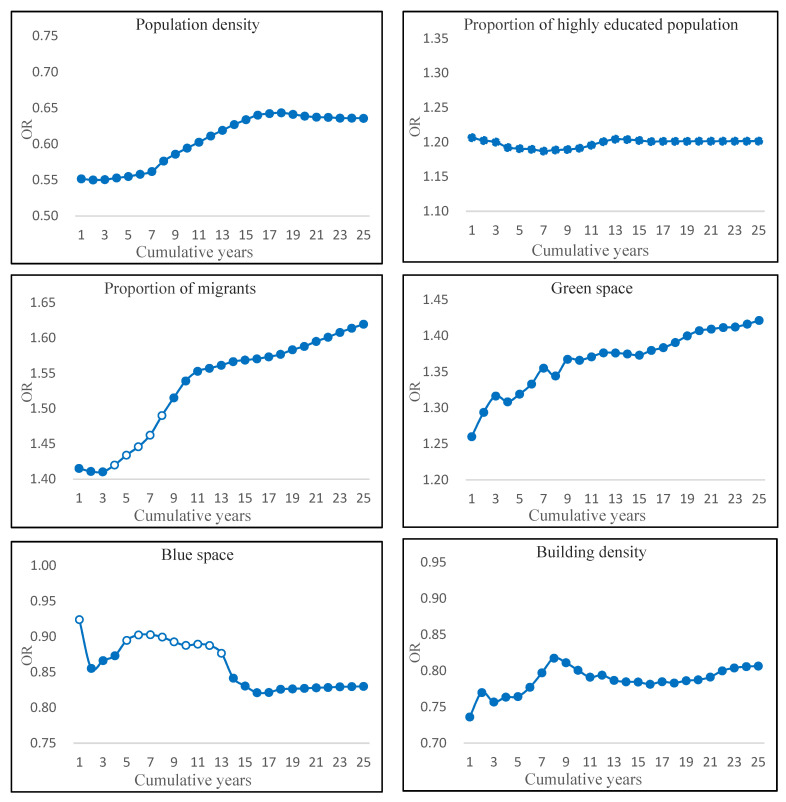
Accumulated exposure to residential environment and subjective wellbeing in later life. Note: Solid circles represent statistical significance at the 90% confidence interval, while hollow circles do not.

**Table 1 ijerph-19-13081-t001:** Sociodemographic attributes of the survey participants (*N* = 782).

Variable	Description	%/Mean	Variable	Description	%/Mean
Subjective wellbeing		3.99	Average age (years)		67
Gender	Male	53.84%	Nature of work unit	State-owned enterprises or public institutions	62.40%
	Female	46.16%		Other	37.60%
Education	Primary school and below	9.97%	Employment before retirement	Full-time job	55.75%
	Middle school	60.10%		Other	44.25%
	High school	21.74%	Individual monthly income (RMB)	<1999	7.42%
	University and above	8.18%		2000–4999	76.73%
Marriage status	Unmarried	6.52%		5000–7999	13.81%
	Married	93.48%		>8000	2.05%
Self-rated health ^1^	Good (4–5)	53.20%	Relocation frequency		1.20
	Medium (3)	37.47%	Residential location	Urban center	54.86%
	Bad (1–2)	9.34%		Center and periphery	26.73%
Hukou ^2^	Non-local	19.18%		Urban periphery	18.41%
	Local	80.82%			

^1^ Self-rated health is measured by a single-item five-option question: “What do you think of your health status compared to your peers?” Higher scores reflect greater levels of health. ^2^ Hukou is a household registration status in China. Migrants, in this paper, refer to residents with non-local hukou. The hukou (household registration) system is regarded as the crucial institutional factor in migrant segregation.

**Table 2 ijerph-19-13081-t002:** The results of Ologit models of long-term residential environment exposure and subjective wellbeing in later life.

	(1)		(2)		(3)	
	β	OR	β	OR	β	OR
Population density			−0.453 ***	0.636		
Proportion of highly educated population			0.184 **	1.202		
Proportion of migrants			0.482 ***	1.620		
Green space					0.352 ***	1.421
Blue space					−0.186 **	0.830
Building density					−0.215 ***	0.807
Gender	0.343 **	1.409	0.348 **	1.416	0.366 **	1.443
Age	0.011	1.011	−0.004	0.996	0.005	1.005
Education	−0.255 ***	0.775	−0.364 ***	0.695	−0.224 ***	0.800
Marital status	1.275 ***	3.579	1.250 ***	3.490	1.243 ***	3.466
Self-rated health	0.496 ***	1.643	0.531 ***	1.701	0.502 ***	1.652
*N*	782		782		782	
Log likelihood	−568.579		−556.059		−555.578	
Pseudo R^2^	0.042		0.064		0.065	

Note: ** *p* < 0.01; *** *p* < 0.001.

**Table 3 ijerph-19-13081-t003:** Moderating effect of relocation frequency.

	(1)		(2)		(3)		(4)		(5)		(6)	
	β	OR	β	OR	β	OR	β	OR	β	OR	β	OR
Relocation frequency	0.265 ***	1.304	0.347 ***	1.415	0.134	1.143	0.395 ***	1.484	0.261 ***	1.298	0.401 ***	1.493
Population density	−0.441 ***	0.643										
Population density * Relocation frequency	0.267 ***	1.306										
Proportion of highly educated population			0.219 **	1.245								
Proportion of highly educated population * Relocation frequency			0.370 ***	1.447								
Proportion of migrants					0.483 ***	1.629						
Proportion of migrants * Relocation frequency					0.104	1.110						
Green space							0.378 ***	1.433				
Green space * Relocation frequency							0.265 ***	1.304				
Blue space									−0.178 **	0.837		
Blue space * Relocation frequency									0.141	1.152		
Building density											−0.190 **	0.827
Building density * Relocation frequency											0.304 ***	1.355
Gender	0.376 **	1.457	0.332 **	1.394	0.410 **	1.507	0.364 **	1.439	0.361 **	1.434	0.387 **	1.473
Age	0.012	1.012	−0.003	0.997	0.016	1.016	0.008	1.008	−0.001	0.999	0.008	1.009
Education	−0.202 **	0.817	−0.327 ***	0.721	−0.212 **	0.809	−0.162 *	0.851	−0.255 ***	0.775	−0.201 **	0.818
Marital status	1.122 ***	3.070	1.173 ***	3.233	1.198 ***	3.313	1.111 ***	3.037	1.190 ***	3.288	1.132 ***	3.100
Self-rated health	0.512 ***	1.668	0.534 ***	1.709	0.539 ***	1.714	0.506 ***	1.659	0.546 ***	1.727	0.512 ***	1.668
N	782		782		782		782		782		782	
Log likelihood	−548.828		−558.925		−553.991		−551.484		−563.662		−557.000	
Pseudo R^2^	0.076		0.059		0.068		0.072		0.051		0.063	

Note: * *p* ≤ 0.05; ** *p* < 0.01; *** *p* < 0.001.

**Table 4 ijerph-19-13081-t004:** Moderating effect of residential location.

	Urban Center		Center and Periphery		Urban Periphery	
	β	OR	Β	OR	β	OR
Population density	−1.107 ***	0.331	−1.346 ***	0.260	0.373 **	1.452
Proportion of highly educated population	0.167	1.182	0.503 **	1.654	0.216	1.241
Proportion of migrants	0.867 ***	2.379	0.971 ***	0.671	−0.898 ***	0.407
Green space	0.672 ***	1.959	0.518 **	1.678	−2.161 ***	0.115
Blue space	−0.223 **	0.800	−0.826 ***	0.438	0.329 *	1.390
Building density	−0.606 ***	0.546	−0.221	0.801	0.776 ***	2.172
*N*	429		209		144	

Note: * *p* ≤ 0.05; ** *p* < 0.01; *** *p* < 0.001.

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
