# Peer review of "Long-Term Residential Environment Exposure and Subjective Wellbeing in Later Life in Guangzhou, China: Moderated by Residential Mobility History"

_ijerph, 2022, doi:10.3390/ijerph192013081_

Round 1
Reviewer 1 Report (Previous Reviewer 1)
Recommendation: Minor revision
In this paper, the author has done some research on subjective well-being. The elderly over 60 years old in Guangzhou, China was selected as the research target to explore the correlation between the elderly's subjective well-being in their later years and various factors. Relatively speaking, there are not many researches on subjective well-being and geographical environment. In addition, this study focuses on specific elderly groups, and discusses the impact of long-term residential environment exposure on subjective well-being in later life, as well as the possible moderating effect of residential mobility history. It is a relatively novel and innovative research. It still exists some problems, although the authors have done many works. I have some comments and suggestions, and would like to see the authors’ response to the following comments and suggestions:
1. Commonly, subjective well-being includes both affective and cognitive components. The affective component mainly includes positive emotions and negative emotions, while the cognitive component mainly refers to the cognitive evaluation of the whole life. In general, both affection and cognition need to be evaluated and scored for a comprehensive measure of subjective well-being. In this paper, the authors seem to score subjective well-being directly from questionnaires. Is this method of acquisition rigorous? Are the scores obtained reliable enough to be used in subsequent studies?
2. In this paper, The data obtained from the questionnaire came from 2016. Although this article is concerned about the impact of long-term residential environment exposure on subjective well-being of the elderly, the timeliness of the data is affected?
3. In part of ‘3.4 Residential Mobility History’. it’s mentioned that sample of elderly people was divided into three types: those who have always lived in the urban center, those who have always lived in the urban periphery, and those who have moved between living in the center and the periphery. It can also be seen from the paper that the built-up area of Guangzhou increased from 206 km2 in 1992 to 1249 km2 in 2016. Whether there are elderly people who have not moved, but the attributes of the area they belong to have changed (For example, the area originally did not belong to the city center, but later belongs to the city center). If this happens, how will such samples be classified?
4. In Table 2 and Table 3, the authors showed the relevant parameters of the models. Such as ‘pseudo R2’ and ‘log likelihood’. Are these parameters sufficient to show the reliability of the model establishment? Is it necessary for authors to add values of other parameters (such as ‘LR chi2(3)’ and ‘Prob > chi2’)?
5. In this paper, the correlation between each factor and subjective well-being is explained and analyzed by means of the coefficient of the variable. Is this explanation sufficient, and whether calculation and analysis of marginal effects are required?
6. In part of ‘4.3 Robustness Check’. It’s mentioned that the type of hukou and the nature of unit before retirement can be used as institutional factors. The authors explained the use of these two factors. However, when using personal monthly income and full-time or part-time employment before retirement as market factors, there is no explanation.
7. The writing of the "6. Conclusion" part of the article is a little lengthy. The author can refine this part and show it better.
Author Response
Dear reviewer:
Thank you for all the helpful comments from you. We have revised our paper carefully to address the comments and have made many corrections, which we hope have adequately addressed your concerns.
In revising the manuscript, we have seriously and carefully considered all the concerns and suggestions you have made. A point-by-point response is provided below for you to easily identify the specific revisions.
Many thanks for your time and kind consideration in this.
Best regards,
Authors
Recommendation: Minor revision
In this paper, the author has done some research on subjective well-being. The elderly over 60 years old in Guangzhou, China was selected as the research target to explore the correlation between the elderly's subjective well-being in their later years and various factors. Relatively speaking, there are not many researches on subjective well-being and geographical environment. In addition, this study focuses on specific elderly groups, and discusses the impact of long-term residential environment exposure on subjective well-being in later life, as well as the possible moderating effect of residential mobility history. It is a relatively novel and innovative research. It still exists some problems, although the authors have done many works. I have some comments and suggestions, and would like to see the authors’ response to the following comments and suggestions:
- Commonly, subjective well-being includes both affective and cognitive components. The affective component mainly includes positive emotions and negative emotions, while the cognitive component mainly refers to the cognitive evaluation of the whole life. In general, both affection and cognition need to be evaluated and scored for a comprehensive measure of subjective well-being. In this paper, the authors seem to score subjective well-being directly from questionnaires. Is this method of acquisition rigorous? Are the scores obtained reliable enough to be used in subsequent studies?
Response: Many thanks for your comment. In recent years, well-being has become a hot topic in national policy and academic research. A large number of studies have shown that subjective well-being can be measured by self-report in questionnaires. Scholars have developed a series of self-report scales based on different conceptual connotations of subjective well-being. There is currently no consensus on the trade-offs between single-item and multi-item scales for measuring subjective well-being. The concise single-item scale is widely used due to its ease of operation. Some studies have confirmed that although the single-item scale cannot well reflect the complexity of subjective well-being, its measurement results are highly correlated with the measurement results of the multi-item scale, indicating that the single-item self-report scale is still scientific and reliable in the research of subjective well-being.
Diener E. Assessing subjective well-being: Progress and opportunities. Social Indicators Research, 1994, 31(2): 103-157.
Diener E, Suh E M, Lucas R E, et al. Subjective well-being: three decades of progress. Psychological Bulletin, 1999, 125(2): 276-302.
Pavot W, Diener E, Colvin C R, et al. Further validation of the Satisfaction with Life Scale: Evidence for the cross-method convergence of well-being measures. Journal of Personality Assessment, 1991, 57(1): 149~161.
Sandvik E, Diener E, Seidlitz L. Subjective Well-Being: The Convergence and Stability of Self-Report and Non-Self-Report Measures. Journal of Personality, 1993, 61(3): 317–342.
Mith T S, Reid L. Which ‘being’ in wellbeing? Ontology, wellness and the geographies of happiness. Progress in Human Geography, 2018, 42(6): 807-829.
Veenhoven R. Happiness: Also Known as “Life Satisfaction” and “Subjective Well-Being”. In: Land K., Michalos A., Sirgy M. (eds) Handbook of Social Indicators and Quality of Life Research. Springer, Dordrecht, 2012. https://doi.org/10.1007/978-94-007-2421-1_3
- In this paper, The data obtained from the questionnaire came from 2016. Although this article is concerned about the impact of long-term residential environment exposure on subjective well-being of the elderly, the timeliness of the data is affected?
Response: It is undeniable that the questionnaire data of 2016 is not sufficiently timely. However, we think that this has little impact on the results of this paper. It can be seen from "5.1" that the relationship between cumulative exposure to residential environment and subjective wellbeing in later life tended to be stable with increase of exposure time. This also confirms that the timeliness of the data will not be affected.
- In part of ‘3.4 Residential Mobility History’. it’s mentioned that sample of elderly people was divided into three types: those who have always lived in the urban center, those who have always lived in the urban periphery, and those who have moved between living in the center and the periphery. It can also be seen from the paper that the built-up area of Guangzhou increased from 206 km2 in 1992 to 1249 km2 in 2016. Whether there are elderly people who have not moved, but the attributes of the area they belong to have changed (For example, the area originally did not belong to the city center, but later belongs to the city center). If this happens, how will such samples be classified?
Response: It is possible that the elderly has not move, but the attributes of the area they belong to have changed. Therefore, in order to solve this problem, this paper takes the outer ring road of Guangzhou as the boundary, and does not include the dynamic changes of the city center boundary into the study. This can avoid the influence of changes of attributes of the area, but may obscure the possible influence of dynamic urban development.
- In Table 2 and Table 3, the authors showed the relevant parameters of the models. Such as ‘pseudo R2’ and ‘log likelihood’. Are these parameters sufficient to show the reliability of the model establishment? Is it necessary for authors to add values of other parameters (such as ‘LR chi2(3)’ and ‘Prob > chi2’)?
Response: We really appreciate your valuable comment. The ordered logistic regression model (Ologit) is used to study the association between long-term residential environment exposure and subjective wellbeing at the individual level. For Ologit,‘pseudo R2’ and ‘log likelihood’ can be independently used to analyze the fitting degree of the model. This study uses two indicators at the same time. These parameters can show the reliability of the model establishment.
- In this paper, the correlation between each factor and subjective well-being is explained and analyzed by means of the coefficient of the variable. Is this explanation sufficient, and whether calculation and analysis of marginal effects are required?
Response: Many thanks for your comment. In this study, β is the correlation coefficient between independent variable and dependent variable. The Odds ratio (OR) indicates the influence of independent variables on the change in the probability of occurrence of dependent variables. These two indicators can fully demonstrate the correlation between long-term residential environment exposure and subjective wellbeing in later life, which is also the focus of this study. We are not able to assess whether there was a causal relationship between geographical context and subjective well-being, as our data were cross-sectional. In future research, calculating and analyzing marginal effects is an important research direction.
- In part of ‘4.3 Robustness Check’. It’s mentioned that the type of hukou and the nature of unit before retirement can be used as institutional factors. The authors explained the use of these two factors. However, when using personal monthly income and full-time or part-time employment before retirement as market factors, there is no explanation.
Response: Because the type of hukou and the nature of unit are the products of a specific period in China and have Chinese characteristics. While personal monthly income and full-time or part-time employment are universal and easier to understand. Therefore, we explain the usage of the type of hukou and the nature of unit in detail. In the revised manuscript, the explanation of using personal monthly income and full-time or part-time employment before retirement as market factors has been added.
“We select the type of hukou and the nature of work unit before retirement as institutional factors, because they are the product of China’s institutional system. Whether residents have local hukou and the associated entitlements and whether they work in a state-owned enterprise or public institution directly affect the quality of residents’ access to public ser-vices such as medical care and education. In addition, personal monthly income and full-time or part-time employment before retirement are selected as market factors, because they are closely related to the market economy.”
(Lines 410-416)
- The writing of the "6. Conclusion" part of the article is a little lengthy. The author can refine this part and show it better.
Response: We really appreciate your helpful suggestion. In the revised manuscript, we have refined the "6. Conclusion" part of the article.
“Taking Guangzhou, China, as a case city, this study explores the relationship be-tween long-term residential environment exposure over 25 years and subjective wellbeing in later life, and the possible moderating effect of residential mobility history. There are three main conclusions. First, subjective wellbeing in later life is not only related to the current residential environment, but also the cumulative exposure to the long-term resi-dential environment. The directions of the relationship between residential environment variables and subjective wellbeing are not affected by cumulative time in the direction (positive or negative), but differ in the intensity. The relationship between long-term residential environment exposure for 15 years or more and subjective wellbeing is relatively stable. Second, the relationship between residential environment and subjective wellbeing is moderated by relocation frequency. Relocation frequency can strengthen the positive relationship between a highly educated population, green space and subjective wellbeing, and weaken the negative relationship between population density, building density and subjective wellbeing. This is consistent with the theory of residential self-selection. Last, the relationship between long-term residential environment exposure and subjective wellbeing in later life is heterogeneous due to the impact of residential location. Although there are differences in the intensity of effects between residential environment and subjective wellbeing of the elderly between the residents who have always lived in the urban center and those who have moved between the center and periphery, the direction of effects (positive effect or negative effect) is relatively consistent. The subjective wellbeing of residents who have always lived in the urban center and those who have always lived in the urban periphery is related to the residential environment in different directions.”
(Lines 484-505)

Reviewer 2 Report (New Reviewer)
The study found that effects of the long-term residential environment on subjective well-being mediated by residential mobility history. Its aims are clear and logics from the issue to conclusion are fine. Although some sentences are redundant, the paper is well structured. However, the authors need to add some explanations on the background of the case study area and rationale to choose the indicators, etc.
Below are specific comments.
p.3, l.119: “Studies have pointed out that long-term environmental exposure is more important than current exposure, because the residential environment may have a continuous impact over time.”
-> The authors need to show several references which pointed out this statement.
p.5, l.195: “A total of 1012 valid questionnaires were collected, and 782 elderly people who have lived in Guangzhou for more than 25 years (since at least 1992) were selected as the sample residents of this study”
-> The authors need to mention the population size of the targets (population of the elderly in Guangzhou, I understand that the number of the elderly staying Guangzhou for more than 25 years might not be accessible to researchers).
-> The authors need to mention how to choose the samples (random? If so, how random?) and how to distribute and collect questionnaires (internet, etc.).
p.5, table 1:
->What do the author mean with “In-system unit” in Nature of work unit? Later, the authors indicate whether they work in a state-owned enterprise or public institution, however, I could not find relations with this.
->As “Employment” means respondents’ work “before retirement,” the authors need to indicate so in the table.
p.5, 3.2 Subjective Wellbeing
And
p.6, 3.3 Exposure Assessment of Residential Environment
And
p.6, 3.4 Residential Mobility History
->Descriptive data should be shown as the authors did in Table 1.
p.6, 3.3 “Exposure Assessment of Residential Environment”
-> The authors need to clearly state that this variable is not only the present living area but it includes their past living areas.
-> The authors need to mention how to calculate this indicator. I could not grasp the frequency of data availability (I guess annual data are available from the latter analysis though there is no indication) and how they are calculated (such as simply accumulated the data of each year: ex. 0.11+0.15+… for the case of population density).
-> Please explain why these specific indicators were chosen. The authors need to mention previous studies or logical explanation why these indicators may be important to subjective well-being (such that how the authors expected NDWI affecting subjective well-being).
p.7, l.256: “In addition, gender, age, education, marital status and self-rated health have been widely discussed in the study of subjective wellbeing, ….”
-> The authors need to show previous studies if they are widely discussed in the previous studies.
p.8, (1) Moderating effect of relocation frequency
-> The authors need to show us its specific indicators. Did the author just count how many times? This comment is the same as the previous one on “p.6, 3.4 Residential Mobility History.”
p.9, l.352: “… based on the median, ….”
-> Please show how much the median is.
p.10, Figure 2
-> There is no alphabetic order, (a), (b), … which is indicated in the main text.
p.12, l.417: “The results show ….”
-> I recommend that the authors add its results like Table 3 in the paper or the appendix.
p.13, 5.2 The Role of Relocation Frequency
-> The authors need to explain us the trend of relocation in the case study area or in the sample. Relocation in this study has four types: moving inside the center areas, moving inside the peripheral areas, moving from the center area to the peripheral area, and moving from the peripheral area to the center area. And each of the moving has different meanings, whereas this study takes them as the same. So is there any major trend of moving or did the result come from the mixture of the moving types? Without answers to this doubt, the discussion of the paper is ambiguous, degrading the quality of the findings.
Author Response
Dear reviewer:
Thank you for all the helpful comments from you. We have revised our paper carefully to address the comments and have made many corrections, which we hope have adequately addressed your concerns.
In revising the manuscript, we have seriously and carefully considered all the concerns and suggestions you have made. A point-by-point response is provided below for you to easily identify the specific revisions.
Many thanks for your time and kind consideration in this.
Best regards,
Authors
The study found that effects of the long-term residential environment on subjective well-being mediated by residential mobility history. Its aims are clear and logics from the issue to conclusion are fine. Although some sentences are redundant, the paper is well structured. However, the authors need to add some explanations on the background of the case study area and rationale to choose the indicators, etc.
Below are specific comments.
p.3, l.119: “Studies have pointed out that long-term environmental exposure is more important than current exposure, because the residential environment may have a continuous impact over time.”
-> The authors need to show several references which pointed out this statement.
Response: Many thanks for your comment. In the revised manuscript, we have provided references to point out this statement.
[10] Morris, T.; Manley, D.; Sabel, C. E. Residential mobility: Towards progress in mobility health research. Prog. Hum. Geogr. 2018, 42, 112-133.
[29] Kyle, A.D.; Woodruff, T.J.; Buffler, P.A.; Davis, D.L. Use of an index to reflect the aggregate burden of long-term exposure to criteria air pollutants in the United States. Environ. Health Perspect. 2002, 110 (Suppl. 1), 95–102.[30] Tzivian, L.; Winkler, A.; Dlugaj, M.; Schikowski, T.; Vossoughi, M.; Fuks, K.; Weinmayr, G.; Hoffmann, B. Effect of long-term outdoor air pollution and noise on cognitive and psychological functions in adults. Int. J. Hyg. Environ. Health 2015, 218, 1–11.
p.5, l.195: “A total of 1012 valid questionnaires were collected, and 782 elderly people who have lived in Guangzhou for more than 25 years (since at least 1992) were selected as the sample residents of this study”
-> The authors need to mention the population size of the targets (population of the elderly in Guangzhou, I understand that the number of the elderly staying Guangzhou for more than 25 years might not be accessible to researchers).
-> The authors need to mention how to choose the samples (random? If so, how random?) and how to distribute and collect questionnaires (internet, etc.).
Response: This study uses data collected from a questionnaire conducted in Guangzhou in 2016. In 2016, there were 1.5461 million elderly people aged 60 and above in Guangzhou, accounting for 17.8% of the total population. The participants in this research were selected on the basis of a stratified random sampling scheme. First of all, based on the Guangzhou census data, this research selected relevant indicators such as population, socio-economic and housing conditions to conduct factor ecological analysis. According to the scores of each factor, the communities in Guangzhou were clustered and grouped into housing community type including high-end commodity housing communities, middle-class communities, affordable housing communities, urban village communities and public housing communities. Typical sample communities with the highest scores of relevant main factors were selected from all the types of communities, and the elderly were randomly surveyed.
A total of 1012 valid questionnaires were collected, and 782 elderly people who have lived in Guangzhou for more than 25 years (since at least 1992) were selected as the sample residents of this study (Table 1). The social and demographic characteristics of participants, their subjective well-being and their history of residential mobility since birth were obtained through a questionnaire. Details of the survey are in Su et al. (2021).
p.5, table 1:
->What do the author mean with “In-system unit” in Nature of work unit? Later, the authors indicate whether they work in a state-owned enterprise or public institution, however, I could not find relations with this.
->As “Employment” means respondents’ work “before retirement,” the authors need to indicate so in the table.
Response: In China, "In system unit" in nature of work unit generally refers to state-owned enterprises and public institutions. For better understanding, we have changed "In-system unit" in Table 1 to "State-owned enterprises or public institutions". In addition, we have changed "Employment" to "Employment before retirement".
p.5, 3.2 Subjective Wellbeing
And
p.6, 3.3 Exposure Assessment of Residential Environment
And
p.6, 3.4 Residential Mobility History
->Descriptive data should be shown as the authors did in Table 1.
Response: Thanks for your comment. In the revised manuscript, we have added subjective wellbeing and residential mobility history to Table 1. The calculation of residential environment variables is complicated, which is explained in detail in the main text.
Table 1. Sociodemographic attributes of the survey participants (N=782)
p.6, 3.3 “Exposure Assessment of Residential Environment”
-> The authors need to clearly state that this variable is not only the present living area but it includes their past living areas.
-> The authors need to mention how to calculate this indicator. I could not grasp the frequency of data availability (I guess annual data are available from the latter analysis though there is no indication) and how they are calculated (such as simply accumulated the data of each year: ex. 0.11+0.15+… for the case of population density).
-> Please explain why these specific indicators were chosen. The authors need to mention previous studies or logical explanation why these indicators may be important to subjective well-being (such that how the authors expected NDWI affecting subjective well-being).
Response: This study focuses on the impact of long-term residential environment exposure on subjective wellbeing in later life. We have considered the dual changes of residential neighborhood and residential environment. The study took the year as the unit and accumulated the residential environment exposure year by year. The selection of indicators is based on both the existing research and the availability of data. In the revised manuscript, we have added relevant descriptions of the above issues.
“3.3 Exposure Assessment of Residential Environment
Residential environment concerned in this study is the social and physical environment of the residential neighborhood. Based on literature review and data availability,……
Assessment of long-term residential environment exposure was based on residential location over the past 25 years. The questionnaire recorded each participant’s residential address and residence time (start/end), which allowed us to accumulate the residential environment variables year by year, so that the dual changes of residence and environment could be taken into account at the same time.”
(Lines 215-244)
p.7, l.256: “In addition, gender, age, education, marital status and self-rated health have been widely discussed in the study of subjective wellbeing, ….”
-> The authors need to show previous studies if they are widely discussed in the previous studies.
Response: The relationship between these individual attribute variables and subjective wellbeing has been widely discussed, but they are not the focus of this study. Therefore, we put these individual attributes into the model as control variables, which are explained and discussed in the model results section.
“In terms of individual socioeconomic attribute variables, women, married people and residents with high self-rated health have higher subjective wellbeing, which is consistent with existing research. However, education status is significantly negatively correlated with subjective wellbeing. This may be because the sample of this study is people over 60 years old, and higher educational attainment does not continue to bring higher competitiveness in the labor market. Highly educated people also tend to have higher expectations for social development and their personal life. All these factors may lead to the decline of their subjective wellbeing.”
(Lines 313-320)
p.8, (1) Moderating effect of relocation frequency
-> The authors need to show us its specific indicators. Did the author just count how many times? This comment is the same as the previous one on “p.6, 3.4 Residential Mobility History.”
Response: Relocation frequency refers to the number of times the sample elderly moved from 1992 to 2016. The average of relocation frequency for the sample is 1.2, which is added in Table 1.
p.9, l.352: “… based on the median, ….”
-> Please show how much the median is.
Response: The median of relocation frequency is 1. Those with relocation frequency of 0 and 1 are in the low relocation frequency group, and those with relocation frequency of 2 and above are in the high relocation frequency group.
p.10, Figure 2
-> There is no alphabetic order, (a), (b), … which is indicated in the main text.
Response: In the revised manuscript, we have added alphabetical order to Figure 2.
p.12, l.417: “The results show ….”
-> I recommend that the authors add its results like Table 3 in the paper or the appendix.
Response: In the revised manuscript, we have added model results of robustness checks in the appendix.
Appendix
Table A. The results of Ologit models of long-term residential environment exposure and subjective wellbeing in later life
(Lines 525-529)
p.13, 5.2 The Role of Relocation Frequency
-> The authors need to explain us the trend of relocation in the case study area or in the sample. Relocation in this study has four types: moving inside the center areas, moving inside the peripheral areas, moving from the center area to the peripheral area, and moving from the peripheral area to the center area. And each of the moving has different meanings, whereas this study takes them as the same. So is there any major trend of moving or did the result come from the mixture of the moving types? Without answers to this doubt, the discussion of the paper is ambiguous, degrading the quality of the findings.
Response: We really appreciate your valuable comment. The relocation frequency and residential location characteristics of the sample elderly in this study have been added to Table 1. We agree with the reviewer that different residential mobility directions may have different impacts. However, this study focuses on the residential history of different locations—the difference in the influence of the geographical environment of the central/peripheral area. If residential mobility is subdivided, it can be divided into the following types: moving inside the center areas, moving inside the peripheral areas, moving from the center area to the peripheral area, moving from the peripheral area to the center area, and multiple round-trip movements between the center area and the peripheral area, etc. This may be an interesting research direction in the future.

Reviewer 3 Report (New Reviewer)
1. In the introduction or any initial part of the article, apart from some general references to the topic, examples of research on a similar subject carried out in other countries and cities, operating under different socio-economic conditions (e.g. from America or Europe) would have been useful. These non-Chinese cases could be a benchmark for the situation in the Chinese city under study. Then, in the final section of the paper, the authors could make a comparison between the findings for the Chinese city and those American or European. This would give readers a broader perspective on the issue at hand, not just limited to the Chinese city.
2. Some puzzlement and doubt is raised by the general question posed to survey respondents: “Taken 209 all together, how happy do you feel about your current life?” Did respondents know what elements to take into account when assessing their wellbeing? If not, each may have included different elements, and this distorts the comparability of the survey results. It is advisable for the authors to briefly address this issue in the article.
3. One other point worth clarifying is why the authors did not include in the study people who moved ( at some stage in their lives) to the city under study from rural areas. After all, such people may manifest a particular perspective on their well-being, broadening the spectrum of opinions on the subject.
Author Response
Dear reviewer:
Thank you for all the helpful comments from you. We have revised our paper carefully to address the comments and have made many corrections, which we hope have adequately addressed your concerns.
In revising the manuscript, we have seriously and carefully considered all the concerns and suggestions you have made. A point-by-point response is provided below for you to easily identify the specific revisions.
Many thanks for your time and kind consideration in this.
Best regards,
Authors
- In the introduction or any initial part of the article, apart from some general references to the topic, examples of research on a similar subject carried out in other countries and cities, operating under different socio-economic conditions (e.g. from America or Europe) would have been useful. These non-Chinese cases could be a benchmark for the situation in the Chinese city under study. Then, in the final section of the paper, the authors could make a comparison between the findings for the Chinese city and those American or European. This would give readers a broader perspective on the issue at hand, not just limited to the Chinese city.
Response: Many thanks for your comment. An important contribution of this study is to verify link between long-term residential environmental exposure and subjective wellbeing in later life. Previous studies mainly focused on the impact of the current residential neighborhood environment. We considered the dual changes of the residential location and neighborhood environment. Of course, our case city is Guangzhou, China. The study area has its distinctive particularity, which is described in the research background. The model results are also interpreted in the context of Chinese cities and dialogues with existing literature. What about the results of the study in American or European cities? Are research findings on Chinese cities applicable to cities in other countries? These issues need further research to explore.
- Some puzzlement and doubt is raised by the general question posed to survey respondents: “Taken 209 all together, how happy do you feel about your current life?” Did respondents know what elements to take into account when assessing their wellbeing? If not, each may have included different elements, and this distorts the comparability of the survey results. It is advisable for the authors to briefly address this issue in the article.
Response: Subjective wellbeing is an individual's self-evaluation of their own life quality. It takes into account the subjective aspects of preference satisfaction and allows people to perceive the quality of their own lives. This study focuses on overall subjective wellbeing, rather than a specific aspect of subjective wellbeing. Subjective wellbeing measurement method used in the research has been widely recognized and applied, and is considered to be scientific, valid and comparable.
“Existing studies have shown single items to measure subjective wellbeing are reliable, effective and comparable [45,46].”
(Lines 212-214)
- One other point worth clarifying is why the authors did not include in the study people who moved (at some stage in their lives) to the city under study from rural areas. After all, such people may manifest a particular perspective on their well-being, broadening the spectrum of opinions on the subject.
Response: We agree with you that those who moved (at some stage in their lives) to the city may manifest a particular perspective on their well-being. Some studies have confirmed the particularity of subjective wellbeing of migrants. However, this study focuses on the impact of long-term residential environment exposure on subjective well-being. In order to control the influence of macro environment, the research sample selected the elderly who have been living in Guangzhou since 1992. In the future, relevant research on people who moved (at some stage in their lives) to the city from rural areas could be strengthened.

Round 2
Reviewer 2 Report (New Reviewer)
The authors revised based on the comments almost sufficiently. Below are two more comments related to the authors' revision/comment.
1. The paper should be self-sufficient, meaning that the paper needs to mention the questionnaire methodologies (not just referring to the previous study) and need to write what kind of random sampling method was adopted (using random number for selection of individual respondents, etc.).
2. About my comment
—
p.7, l.256: “In addition, gender, age, education, marital status and self-rated health have been widely discussed in the study of subjective wellbeing, ….”
-> The authors need to show previous studies if they are widely discussed in the previous studies.
—
What I meant was that even though it is not a focus of this study, if the authors mention previous studies, they need to show some examples of the previous studies. Otherwise, readers cannot find the credibility of this sentence.
Author Response
The authors revised based on the comments almost sufficiently. Below are two more comments related to the authors' revision/comment.
1.The paper should be self-sufficient, meaning that the paper needs to mention the questionnaire methodologies (not just referring to the previous study) and need to write what kind of random sampling method was adopted (using random number for selection of individual respondents, etc.)
Response: Many thanks for your suggestion. This study and another published paper of the authors used the same questionnaire. In the revised manuscript, we have added the questionnaire methodologies. The study area and the questionnaire survey were introduced in more detail in Su et al. (2021).
“We adopted stratified random sampling survey method. Firstly, based on the data of the sixth Census of Guangzhou, the neighborhoods were clustered, and then a total of 46 neighborhoods were selected from various types of neighborhoods as sample neighborhoods to conduct the random sampling survey of the elderly. More details of the survey are in Su et al. (2021) [42].”
(Lines 195-199)
- About my comment
—
p.7, l.256: “In addition, gender, age, education, marital status and self-rated health have been widely discussed in the study of subjective wellbeing, ….”
-> The authors need to show previous studies if they are widely discussed in the previous studies.
—
What I meant was that even though it is not a focus of this study, if the authors mention previous studies, they need to show some examples of the previous studies. Otherwise, readers cannot find the credibility of this sentence.
Response: We really appreciate your valuable comment. In the revised manuscript, we have provided references to increase the credibility of this sentence.
[52] Ha, S.; Kim, S. Personality and subjective well-being: evidence from South Korea. Social Indic. Res. 2013, 111, 341–359.
[53] Goswami, H. Children's subjective well-being: socio-demographic characteristics and personality. Child Indic. Res. 2014, 7, 119–140.
This manuscript is a resubmission of an earlier submission. The following is a list of the peer review reports and author responses from that submission.
Round 1
Reviewer 1 Report
The authors of this paper describe the impact of long-term residential environment on subjective well-being in people aged 60 years and over in Guangzhou, China. This study clarifies that subjective well-being in later life is not only related to the current residential environment but also influenced by the long-term residential environment, and further analyses the effects of relocation frequency and residential location on subjective well-being. The authors have done a lot of related data collection and analysis work. However, many problems still need to be solved. I think it can be accepted and published after solving some questions below:
1. In the Abstract, the authors present the study can provide guidance for individuals’ residential choice and governance of the urban environment to improve wellbeing. Please clarify how this study provides guidance for the governance of urban environments to improve well-being.
2. The introduction only cites 6 references, I think the number of references cited should be increased.
3. In the Introduction, the introduction only introduces the concept and lacks a description of current research progress, please add to the existing research progress or the research results of others to enrich the integrity of the introduction.
4. In line 166, 2. should be 3.
5. In the Data and Methods, please clarify how the NDVI, NDWI and NDBI were obtained through band calculation. Please elaborate on the role of these in the text.
6. In the Result, please clarify how these values(β, OR , p) were obtained.
7. The article mentions that living in a green environment for a long time increases happiness in old age, but it seems to me that this conclusion is well known even without research, which is similar to common sense. Please add more novel conclusions to add and explore.
8. The conclusions of this article only express the author's conclusions from the research data and are not supported by the literature of others, which in my opinion does not contribute to the reliability of the conclusions of the article. Please add to the conclusion section a discussion of existing studies that can support the conclusions of this article to increase the credibility of the conclusions.
9. Chart formatting errors: (1) There is an obvious error in the format of the table. For example, the numbers and punctuation on different lines. (2) Figure 2., Figure 3. and Figure 4 should use the same font. (3) Formula numbers should be right-justified.
10. In the References, the references have two serial numbers.
11. Author's contribution not filled in, please complete.
Author Response
The authors of this paper describe the impact of long-term residential environment on subjective well-being in people aged 60 years and over in Guangzhou, China. This study clarifies that subjective well-being in later life is not only related to the current residential environment but also influenced by the long-term residential environment, and further analyses the effects of relocation frequency and residential location on subjective well-being. The authors have done a lot of related data collection and analysis work. However, many problems still need to be solved. I think it can be accepted and published after solving some questions below:
- In the Abstract, the authors present the study can provide guidance for individuals’ residential choice and governance of the urban environment to improve wellbeing. Please clarify how this study provides guidance for the governance of urban environments to improve well-being.
Response: Many thanks for your comment. In the revised manuscript, we have supplemented the contents of this aspect. Our study further confirms that subjective wellbeing in later life is not only related to current residential environment exposure, but also to long-term residential environment cumulative exposure. Only focusing on the current residential neighborhood environment may overestimate or underestimate the impact of geographical environment on subjective wellbeing. Appropriate residential mobility can improve residents’ subjective wellbeing. Policy makers and urban managers need to improve and standardize the housing supply system to meet the housing needs of different socioeconomic groups and promote reasonable and healthy residential mobility. In future urban planning and management, it is necessary to plan according to the functions of different spaces and the needs of the space users, to strengthen the livability and sustainability of cities.
- The introduction only cites 6 references, I think the number of references cited should be increased.
Response: In the revised manuscript, we have increased the number of references cited.
Bai, X.; Nath, I.; Capon, A.; Hasan, N.; Jaron, D. Health and wellbeing in the changing urban environment: complex challenges, scientific responses, and the way forward. Curr. Opin. Environ. Sustain. 2012, 4, 465-472.
Morrison, P. S. Local expressions of subjective well-being: The New Zealand experience. Reg. Stud. 2011, 45, 1039–1058.
Douglas, O.; Lennon, M.; Scott, M. Green space benefits for health and well-being: a life-course approach for urban planning, design and management. Cities 2017, 66, 53-62.
Ballas, D.; Tranmer, M. Happy people or happy places? A multilevel modeling approach to the analysis of happiness and well-being. Int. Reg. Sci. Rev. 2012, 35, 70-102.
Eid, M.; Diener, E. Global judgments of subjective well-being: situational variability and long-term stability. Soc. Indic. Res. 2004, 65, 245-277.
Morris, T.; Manley, D.; Sabel, C. E. Residential mobility: Towards progress in mobility health research. Prog. Hum. Geogr. 2018, 42, 112-133.
Coulter, R.; van Ham, M.; Findlay, A. M. Re-thinking residential mobility: linking lives through time and space. Prog. Hum. Geogr. 2016, 40, 352-374.
Park, Y. M.; Kwan, M. P. Individual exposure estimates may be erroneous when spatiotemporal variability of air pollution and human mobility are ignored. Health Place 2017, 43, 85-94.
Li, S.; Mao, S. The spatial pattern of residential mobility in Guangzhou, China. Int. J. Urban Reg. Res. 2019, 43(5), 963–982.
Van Kamp, I.; Leidelmeijer, K.; Marsman, G.; De Hollander, A. Urban environmental quality and human well-being. Landsc. Urban Plan. 2006, 65, 5-18.
- In the Introduction, the introduction only introduces the concept and lacks a description of current research progress, please add to the existing research progress or the research results of others to enrich the integrity of the introduction.
Response: Many thanks for your comment. We have added more descriptions of current research progress to enrich the integrity of the introduction.
“The value of linking urban environment and wellbeing outcomes is being increasingly recognized, however the myriad relationships are far from being understood scientifically [5,6]. The effects of residential environment exposure on subjective well-being in multiple temporal and spatial dimensions are conditioned by complicated interactions [7]. Both the duration and spatial location of environmental exposure may lead to different subjective well-being outcomes [8,9]. Besides, residents’ living conditions and environment have persistent differences, which has severe effects on individuals’ subjective well-being. To a certain extent, residential mobility is the major cause of continuous population differentiation in environmental health and well-being [10]. Existing studies tend to focus on the impact of the current normalized residential neighbor-hood environment on subjective well-being, but for most residents, the residential environment is not static, especially in recent years, residential mobility has become more and more common [11]. Subjective well-being as an overall assessment of a person's long-term quality of life. It may be important to analyze the relationship between long-term residential environment exposure and subjective well-being in the context of residential mobility.”
(Lines 46-60)
- In line 166, 2. should be 3.
Response: In the revised manuscript, we have revised it.
- In the Data and Methods, please clarify how the NDVI, NDWI and NDBI were obtained through band calculation. Please elaborate on the role of these in the text.
Response: We really appreciate your valuable comment. In the Data and Methods, we have added descriptions of NDVI, NDWI and NDBI. There are mature and widely accepted calculation methods for these three indicators, see references Tucker (1979), McFeeters (1996) and Zha et al. (2003).
The arithmetic calculation of bands: red (Red), green (Green), near-infrared (NIR) and mid-infrared (MIR).
Tucker, C. J. Red and photographic infrared linear combinations for monitoring vegetation. Remote Sens. Environ. 1979, 8, 127-150
McFeeters, S. K. The use of the Normalized Difference Water Index (NDWI) in the delineation of open water features. Int. J. Remote Sens. 1996, 17, 1425-1432.
Zha, Y.; Gao, Y.; Ni, S. Use of normalized difference built-up index in automatically mapping urban areas from TM imagery. Int. J. Remote Sens. 2003, 24, 583–594.
“To represent the residential physical environment, the normalized difference vegetation index (NDVI), the normalized difference water index (NDWI) and the normalized difference built-up index (NDBI) were obtained through band calculation. NDVI is a numerical indicator that uses the visible and near-infrared bands of the electromagnetic spectrum, which can effectively capture neighborhood green space [44]. NDWI has been developed to extract open water features or blue space and enhance their presence in remote sensing images based on reflected near-infrared radiation and visible green light [45]. NDBI uses the difference in reflection characteristics of buildings in mid-infrared and near-infrared to discriminate impervious surfaces, and then extract urban building density [46]. The physical environment variables in this study were obtained based on the 1km buffer dis-tance surrounding residential location, as this distance has been found to be efficacious in health and wellbeing studies employing multiple buffer distances [47,48].”
(Lines 227-239)
- In the Result, please clarify how these values (β, OR, p) were obtained.
Response: The dependent variable is the current subjective wellbeing, which is a five-category ordinal variable, so the ordered logistic regression model (Ologit) is used to study the as-sociation between long-term residential environment exposure and subjective wellbeing at the individual level. β is the regression coefficient. The Odds ratio (OR) indicates the influence of independent variables on the change in the probability of occurrence of dependent variables. P represents the significance level at 95% confidence interval.
- The article mentions that living in a green environment for a long time increases happiness in old age, but it seems to me that this conclusion is well known even without research, which is similar to common sense. Please add more novel conclusions to add and explore.
Response: Many thanks for your comment. One of our findings is that the elderly with long-term exposure to green space have higher subjective wellbeing. At the same time, it is found that the positive effect of green space is enhanced with the increase of cumulative years. Green space has a higher correlation with subjective wellbeing in older adults with the history of frequent relocation. At the same time, it is found that green space has different effects on subjective wellbeing in the central and peripheral areas of the city. This paper incorporates residential mobility into the research framework of environment and subjective wellbeing, which enriches the research on long-term residential environment exposure and subjective wellbeing in later life.
- The conclusions of this article only express the author's conclusions from the research data and are not supported by the literature of others, which in my opinion does not contribute to the reliability of the conclusions of the article. Please add to the conclusion section a discussion of existing studies that can support the conclusions of this article to increase the credibility of the conclusions.
Response: We really appreciate your valuable comment. In the interpretation of the model results, the support of relevant literature has been added. In the discussion section, the discussion of existing studies has been added to support the conclusions of this article to increase the credibility of the conclusions. Please see the full text of the revised manuscript for details.
- Chart formatting errors: (1) There is an obvious error in the format of the table. For example, the numbers and punctuation on different lines. (2) Figure 2., Figure 3. and Figure 4 should use the same font. (3) Formula numbers should be right-justified.
Response: In the revised manuscript, we have revised it.
- In the References, the references have two serial numbers.
Response: In the revised manuscript, we have corrected this error.
- Author's contribution not filled in, please complete.
Response: We really appreciate your helpful suggestion. In the revised manuscript, we have completed the author's contribution.

Reviewer 2 Report
The research topic of this manuscript is not novel enough. Further, the main concerns of the reviewer are as follows:
1. The full text of the manuscript can hardly find a clear definition of the so called term “Long-term Residential Environment Exposure”, making it difficult to read and understand.
2. The study area is not representative since Guangzhou is an economically developed city. As is known to all, economic level has a significant impact on subjective well-being.
3. Subjective well-being is a potential variable, and it is not convincing to use a simple observation variable to reflect it.
4. I suggest that the manuscript structure should be organized in the form of hypothesis testing.
5. It is inappropriate for China to be the keyword of the paper. Also, the manuscript did not explain the value of China’s case study.
6. The manuscript is not arranged according to the template of the journal, including title format and reference format.
7. In terms of language, this manuscript presents a deficit in the quality of argumentation.
On the whole, in my opinion, this manuscript has little value for publication. I sincerely suggest the authors to start with the theoretical basis and explore the interesting research topics, rather than focusing on some meaningless analysis.
Author Response
The research topic of this manuscript is not novel enough. Further, the main concerns of the reviewer are as follows:
- The full text of the manuscript can hardly find a clear definition of the so called term “Long-term Residential Environment Exposure”, making it difficult to read and understand.
Response: We really appreciate your valuable comment. In the revised manuscript, we have added explanations. Residential environment concerned in this study is the social and physical environment of the residential neighborhood, and accumulate the residential environment variables year by year.
“Residential environment concerned in this study is the social and physical environ-ment of the residential neighborhood. Based on literature review and data availability, the population density, the proportion of migrants and a highly educated population (undergraduate and above) in the neighborhood are used as proxies for residential social environment……To represent the residential physical environment, the normalized difference vegetation index (NDVI), the normalized difference water index (NDWI) and the normalized difference built-up index (NDBI) were obtained through band calculation……
Assessment of long-term residential environment exposure was based on residential location over the past 25 years. The questionnaire recorded each participant’s residential address and residence time (start/end), which allowed us to accumulate the residential environment variables year by year, so that the dual changes of residence and environment could be taken into account at the same time.”
(Lines 216 -244)
- The study area is not representative since Guangzhou is an economically developed city. As is known to all, economic level has a significant impact on subjective well-being.
Response: We agree with you that economic level has a significant impact on subjective well-being. Participants in this study come from the same city and have a common urban economic environment background, and on this basis to analyze the impact of long-term residential environment exposure on subjective wellbeing in later life. At the same time, the individual monthly income was included in the robustness text, and the results showed that income did not change the relationship between residential environment exposure and subjective wellbeing in later life. Guangzhou is at the forefront of China’s market economy transformation, and the urban environment is changing rapidly. Guangzhou has the common characteristics of Chinese cities, but also has its own uniqueness. We believe that it is interesting and meaningful to study the relationship between long-term residential environment exposure and subjective wellbeing in Guangzhou.
- Subjective well-being is a potential variable, and it is not convincing to use a simple observation variable to reflect it.
Response: In recent years, well-being has become a hot topic in national policy and academic research. A large number of studies have shown that subjective well-being can be measured by self-report in questionnaires. Scholars have developed a series of self-report scales based on different conceptual connotations of subjective well-being. There is currently no consensus on the trade-offs between single-item and multi-item scales for measuring subjective well-being. The concise single-item scale is widely used due to its ease of operation. Some studies have confirmed that although the single-item scale cannot well reflect the complexity of subjective well-being, its measurement results are highly correlated with the measurement results of the multi-item scale, indicating that the single-item self-report scale is still scientific and reliable in the research of subjective well-being.
Diener E. Assessing subjective well-being: Progress and opportunities. Social Indicators Research, 1994, 31(2): 103-157.
Diener E, Suh E M, Lucas R E, et al. Subjective well-being: three decades of progress. Psychological Bulletin, 1999, 125(2): 276-302.
Pavot W, Diener E, Colvin C R, et al. Further validation of the Satisfaction with Life Scale: Evidence for the cross-method convergence of well-being measures. Journal of Personality Assessment, 1991, 57(1): 149~161.
Sandvik E, Diener E, Seidlitz L. Subjective Well-Being: The Convergence and Stability of Self-Report and Non-Self-Report Measures. Journal of Personality, 1993, 61(3): 317–342.
Mith T S, Reid L. Which ‘being’ in wellbeing? Ontology, wellness and the geographies of happiness. Progress in Human Geography, 2018, 42(6): 807-829.
Veenhoven R. Happiness: Also Known as “Life Satisfaction” and “Subjective Well-Being”. In: Land K., Michalos A., Sirgy M. (eds) Handbook of Social Indicators and Quality of Life Research. Springer, Dordrecht, 2012. https://doi.org/10.1007/978-94-007-2421-1_3
- I suggest that the manuscript structure should be organized in the form of hypothesis testing.
Response: We really appreciate your helpful suggestion. In the revised manuscript, we have adopted the form of hypothesis testing.
“2.4 Conceptual Framework and Research Hypothesis
Based on the analysis and review above, this paper proposes a conceptual frame-work to examine and compare the relationship of long-term residential environment exposure and subjective wellbeing under residential mobility (Figure 1). There are two hypotheses.
Hypothesis 1. Subjective wellbeing in later life is related to the long-term residential environment exposure.
Hypothesis 2. This relationship may be moderated by residential mobility history such as relocation frequency and residential location.”
(Lines 173-180)
- It is inappropriate for China to be the keyword of the paper. Also, the manuscript did not explain the value of China’s case study.
Response: Compared with cities in Western countries, Chinese cities have a special era background, which have experienced the transition from planned economy to market economy. In the process of this transformation, residential mobility become more and more common, and the urban environment has undergone significant changes. Explained in “2.1 Institutional Reform and Residential Mobility”. The research background has certain Chinese characteristics, and the research results are also explained in the context of Chinese cities. Whether the research conclusions are applicable to cities in Western countries needs to be further verified.
- The manuscript is not arranged according to the template of the journal, including title format and reference format.
Response: In the revised manuscript, we have arranged according to the format requirements of this journal.
- In terms of language, this manuscript presents a deficit in the quality of argumentation.
Response: Many thanks for your comment. We have proofread the manuscript thoroughly. Please see the full text of the revised manuscript for details.
